# Transformations in Exposure to Debris Flows in Post-Earthquake Sichuan, China

Isabelle Utley[1] , Tristram Hales[1] , Ekbal Hussain[2] , Xuanmei Fan[3]

[1] School of Earth and Environmental Sciences, Cardiff University, Cardiff, CF10 3AT, UK.
[2] British Geological Survey, Keyworth, Nottingham, NG12 5GG, UK
[3] State Key laboratory of Geohazard Prevention, Chengdu University of Technology, Chengdu, China

*Correspondence to: Isabelle E.K Utley (utleyieu@gmail.com)*

**Abstract.** Post-earthquake debris flows can exceed volumes of $1 \times 10^6 m^3$ and pose significant challenges to downslope recovery zones. These stochastic hazards form when intense rain remobilises coseismic landslide material. As communities recover from earthquakes they mitigate the effects of these debris flows through modifications to catchments such as building check dams and levees. We investigate how different catchment interventions change exposure and hazard of post-2008 debris flows in three gullies in Sichuan province, China. These were selected based on the number of post-earthquake check dams – Cutou (2), Chediguan (2) and Xiaojia (none). Using high resolution satellite images, we developed a multitemporal building inventory from 2005 to 2019, comparing it to spatial distribution of previous debris flows and future modelled events. Post-earthquake urban development in Cutou and Chediguan increased exposure to a major debris flow in 2019 with inundation impacting 40% and 7% of surveyed structures respectively. We simulated future debris flow runouts using LAHARZ to investigate the role of check dams in mitigating three flow volumes – $10^4 m^3$ (low), $10^5 m^3$ (high) and $10^6 m^3$ (extreme). Our simulations show check dams effectively mitigate exposure to low and high flow events but prove ineffective for extreme events with 59% of buildings in Cutou, 22% in Chediguan and 33% in Xiaojia significantly affected. We verified our analyses through employing a statistical exposure model, adapted from a social vulnerability equation. Cutou's exposure increased by 64% in 2019, Chediguan's by 52% whilst only 2% for Xiaojia in 2011, highlighting that extensive grey infrastructure correlates with higher exposure to extreme debris flows, but less so to smaller events. Our work suggests that the presence of check dams contributes to a perceived reduction in downstream exposure. However, this perception can lead to a levee effect, whereby exposure to larger, less frequent events is ultimately increased.

**Keywords**
Debris Flows, Built Environment, Exposure, Check dams, LAHARZ.

## 1. Introduction

Major earthquakes such as the 1994 $M_w$ 6.8 event in Northridge, California (Harp and Jibson, 1996) and the 1999 $M_w$ 7.3 earthquake in Chi-Chi, Taiwan (Liu et al., 2008) have triggered chains of hazards that increase the exposure of local communities to secondary hazards for many years after the initial disaster. Following the 2008 $M_w$ 7.9 Wenchuan earthquake in Sichuan, China, debris flows occurred more frequently and at a higher magnitude ($>1 \times 10^6 m^3$) after the earthquake compared to flows before the earthquake (Cruden and Varnes., 1996; Cui et al., 2008; Huang and Li., 2009; Guo et al., 2016; Thouret et al., 2020). Increased debris flow frequency impacts vulnerable communities and local infrastructure, potentially reshaping the demographic and structural landscape of previously rural regions (Chen et al., 2011). The frequency of post-seismic flows is heavily influenced by sediment availability, often controlled by coseismic landslide distribution, hydrology, and slope (Horton et al., 2019). The ready transformation and remobilisation of seismically loosened deposits into water-laden sediments leads to a heightened probability of debris flow hazards for extended periods, further exacerbating the potential impacts felt by these areas (Costa et al., 1984; Huang & Li, 2014; Fan et al., 2019b).

Post-seismic debris flows affect the expanding built environment and communities located in the flat land that forms along floodplains and on debris and alluvial fans. In addition to direct loss of life, debris flows repeatedly block and/or destroy rivers, roads, tunnels, and bridges, and damage property and agriculture, and result in loss of life (Chen, N et al., 2011). Buildings are particularly susceptible to the impacts of debris flows (Hu et al., 2012; Zeng et al., 2015), with property damage accounting for nearly all impacts such as casualties and fatalities (Wei et al., 2018; Wei et al., 2022). Variations in construction materials are a particularly important factor in determining structural resilience and vulnerability to debris flows (Zhang, S et al., 2018). Despite focus on building resilience and reducing vulnerability, post-earthquake regions are often areas of significant rebuilding

and expansion of infrastructure so the exposure to debris flows changes rapidly in these areas. The development of critical infrastructure such as highways and tunnels further encourages the growth of the built environment and subsequent influx of people settling in areas exposed to geological hazards (Cruden and Varnes, 1996; Jiang et al., 2016).

Check dams are a common form of risk mitigation for debris flows globally (Zeng et al., 2009; Peng et al., 2014; Cucchiaro, S. et al., 2019b), and one that is prevalent in post-earthquake Wenchuan (Chen, X. et al., 2015; Guo et al., 2016). Check dams store debris flow sediment, locally reduce channel slope, and are often permeable to affect debris flow hydrology. However, they have disadvantages such as requiring regular maintenance (to reduce sediment inputs) (Kean et al., 2018). The mitigation potential of these structures is contingent on their position along a channel, their height, amount of sediment fill, and their strength (which depends on the materials used for construction) (Dai et al., 2017). These factors evolve through time, meaning that the hazard-mitigating factor of check dams can vary with time and often with unpredictable results. The presence of check dams changes the downstream risk, primarily by altering the magnitude and frequency distribution of debris flows within the channel. For well-made check dams of sufficient volume to mitigate the largest debris flows, this can reduce the downstream risk of debris flows to negligible by effectively mitigating the entire hazard. However, in the case of the Wenchuan region, check dams are rarely large enough or regularly cleared of sediment to mitigate the largest debris flows, which can exceed $10^6$ m$^3$ in volume.

The presence of check dams, particularly in drainage basins with a limited history of catastrophic debris flow events, may affect the perception of risk downstream. They serve to stabilize, obstruct, drain, and/or halt the movement of flows (Hübl and Fiebiger., 2005; Chen et al., 2015). The perception that check dams have mitigated all hazards may promote the expansion of infrastructure into floodplains and debris fans, potentially increasing exposure to debris flows that overtop dams or occur due to dam failure. The increase of exposure is common on floodplains where the presence of flood control levees can promote building onto floodplains – a process known as the levee effect (Collenteur et al., 2015). In the flooding example, the presence of levees reduces the frequency of small and medium sized floods, but when large floods occur that cause those levees to fail, heightened floodplain exposure can lead to higher damage, The effect of check dams on risk perception is less well understood. Anecdotal examples from the Wenchuan region (e.g.,, Hongchun, Taoguan gullies) show that large debris flow events in 2010, 2013 and 2019 caused significant damage despite the presence of check dams (Dai et al., 2017). However, it is not clear if the presence of check dams affected exposure relative to the large-scale expansion of infrastructure in the post-earthquake recovery phase.

This study seeks to understand whether the addition of engineered mitigation measures, primarily check dams, have influenced the susceptibility of post-earthquake Wenchuan communities to large debris flows. We compare 3 catchments with similar topography and geology, but different levels of mitigation. We measure the building exposure in two neighbouring catchments with check dams (Cutou and Chediguan) and compare with a third, unmitigated gully (Xiaojia). We examine how infrastructure develops in the basins with time and as a function of check dam measures. By analysing infrastructure development in these catchments, particularly in Cutou and Chediguan in the years following mitigation – will seek to assess how check dam construction has impacted infrastructure growth and the potential exposure to debris flow events of different sizes. Additionally, our analysis will explore whether the presence of these structures has impacted risk perception and/or land-use decisions in 'at-risk' catchments.

## 2. Study Area: Sichuan Province, China

China's mountainous regions, including the Longmenshan, account for 69% of the country's land mass with over a third of the population living in these regions (Chen et al., 2011; He et al., 2022). 72% of this landscape suffers from debris flow activity. Between 2005 and 2018, estimates suggest over 800 debris flow occurrences each year (He et al., 2022; Wei et al., 2021) Following the 2008 Wenchuan Earthquake, landslides were widely recorded across both northern and southern provinces, with debris flows particularly concentrated in the steep terrain of southern Sichuan. However, debris flows, and other landslide types have since been documented across a broad range of regions in China (Liu et al., 2018). The 2008 $M_w$ 7.9 Wenchuan earthquake primarily impacted Sichuan province (Fig 1). The epicentre was located near Yinxiu, Wenchuan County, within the seismically active Longmenshan Fault Zone (Li et al., 2019). The shaking triggered around 56,000 landslides and displaced nearly 3 km$^3$ of loose material (Fan et al., 2018; Luo et al, 2020). In subsequent years, the unstable material has been reactivated as debris flows, many of which exceed -$10^6$ m$^3$ in mobilised volume (Frances et al., 2022). The risk from these debris flows has been compounded by increasing exposure due to China's rapid rural development programme, which includes the construction of roads, bridges, and industrial facilities (Tang et al., 2022).

Four significant episodes of debris flows occurred in the post-earthquake Wenchuan region in 2008, 2010, 2013 and 2019 (Tang et al., 2022; Fan et al., 2019b). Each event was associated with monsoon rainfall that occurred in different parts of the range. The largest flow surges, containing millions of cubic meters of sediment were located in the gullies along the Minjiang in Sichuan. Large scale flooding further amplified the impacts, for example in Yingxiu Town, Wenchuan County (Liu et al., 2016b). Debris flow events occurring post-earthquake often exhibit larger material volumes compared to flow events recorded prior to 2008. Horton et al., (2019) attributed the increase in flow volume too high in channel sediment volumes that can drive bulking. The resulting increase in debris flow hazards necessitated engineered mitigation measures to reduce risk levels in the basin communities (Tang et al., 2009; Huang et al., 2009; Huang, 2012).

In this study we focus on three gullies along the Minjiang - Cutou, Chediguan and Xiaojia and debris flow events on August 20$^{th}$, 2019, and 4$^{th}$ July 2011 (Fig 1). Cumulative rainfall on 20$^{th}$ August 2019 peaked at 83 mm in Cutou and 65 mm in Chediguan resulting in large debris flows measuring over $50 \times 10^4$ m$^3$ in each gully. Cutou gully is known for its high frequency of post-seismic debris flows, which has been attributed to the total of $11 \times 10^6$ m$^3$ coseismic deposits generated by the earthquake (Yan et al 2014). Although a check dam was built in 2011 to manage debris flow impacts in Chediguan gully a large damaging debris flow of $64 \times 10^4$ m$^3$ occurred on 20 August 2019 and destroyed the drainage groove and G213 Taiping Middle Bridge (Li et al., 2021). The debris briefly blocked river flow in the Minjiang causing water levels to rise during flood peak. This led to flooding at the Taipingyi hydropower station located 200 m upstream.

Xiaojia gully, is a moderate debris flow hazard area based on limited past occurrences and has no existing engineered mitigation measures. Following a period of debris flow activity in 2010, and after a period of continuous heavy rainfall approximately 30,000m$^3$ of deposits were remobilised and transported along the channel to the gully mouth. This event led to a period of disruption on the S303 road from flooding (Liu et al., 2014).

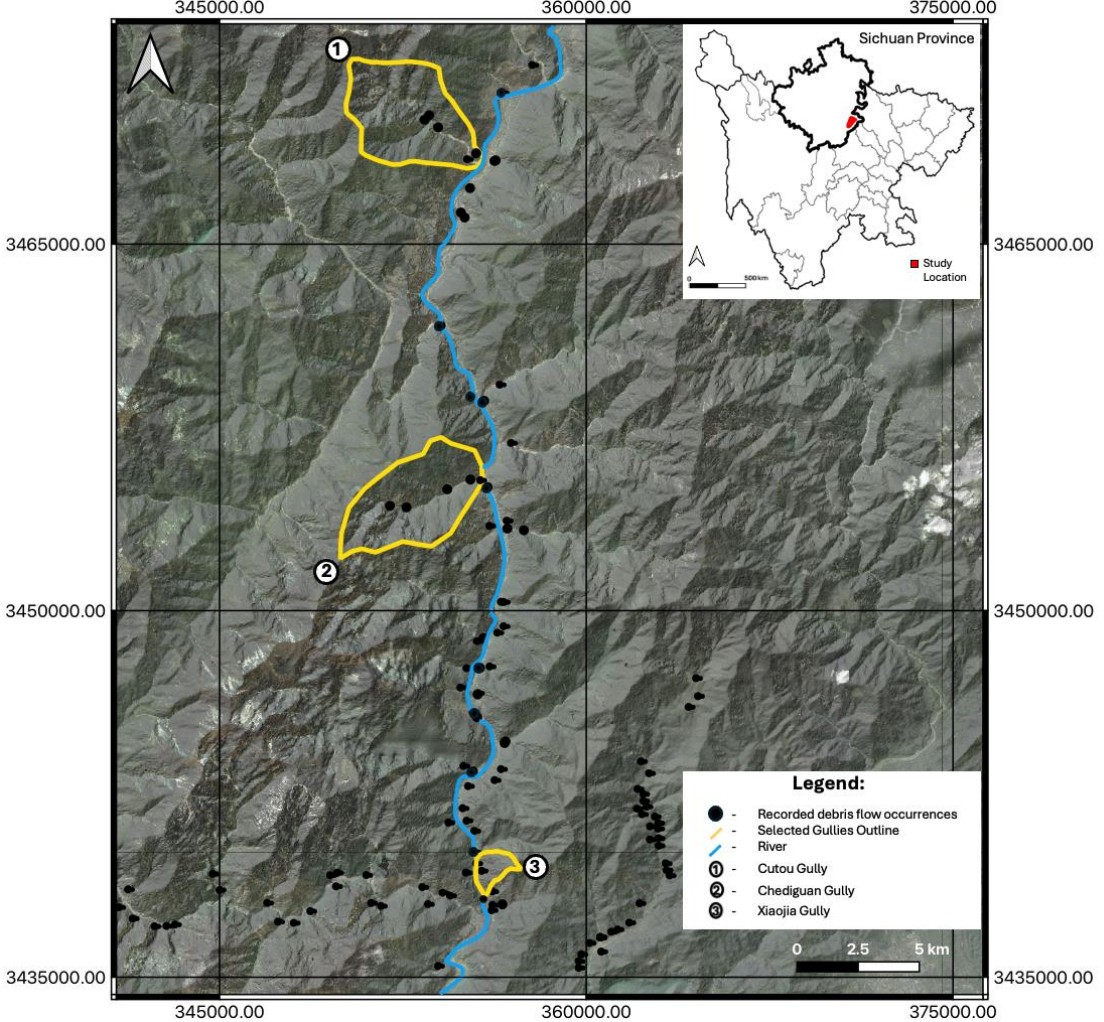

**Figure 1:** Location of the three gullies that form the focus of this study within Sichuan Province. Recorded post-
2008 landslide occurrences are from the Wang et al. (2022) multitemporal datasets (© Google Earth 2019).
**3. Methodology**

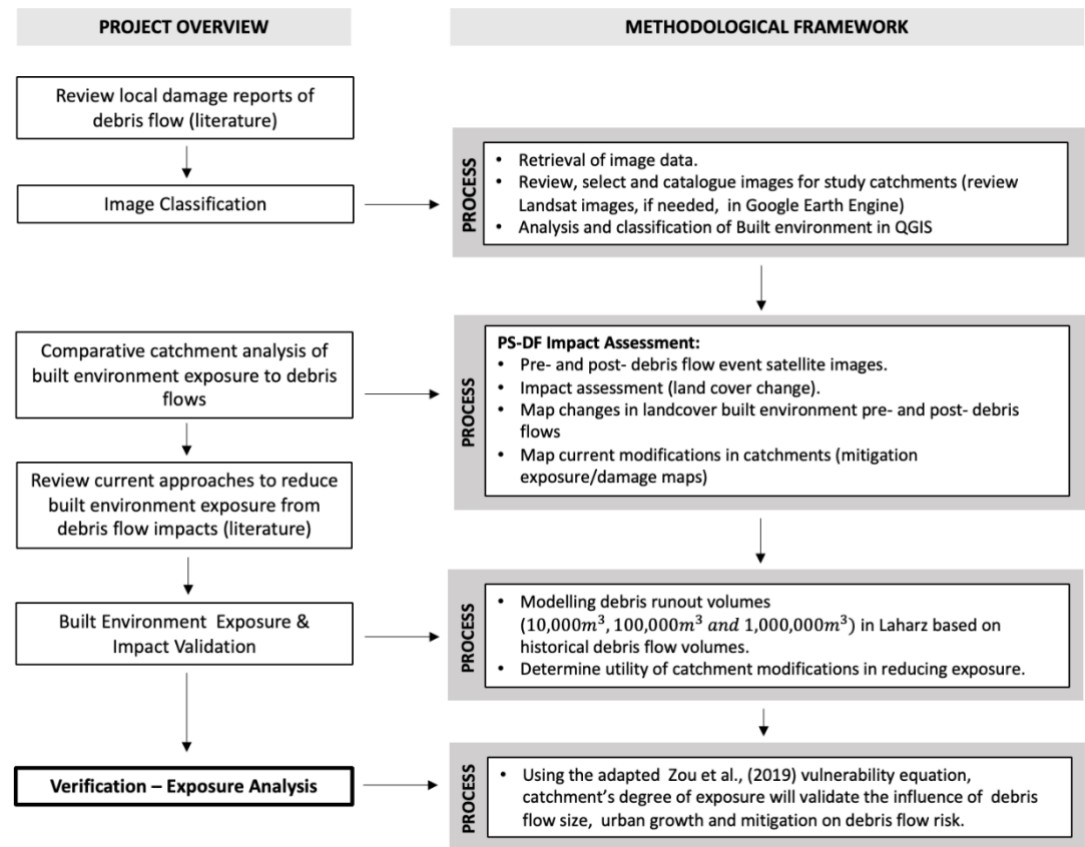

**Figure 2:** Schematic of our method. The key data sources comprise three multi-temporal datasets, including two
from Fan et al. (2019a) covering debris flow and triggering rainfalls, as well as mitigation measures. The third
dataset is adapted from Fan et al., (2019a) and highlights gully's with debris flow events post-2008 including
information on the flow volume and presence of mitigation. Additional spatial data sources include aerial imagery
from OpenStreetMap (OpenStreetMap contributors., 2023), World Settlement Footprint (World Settlement
Footprint., 2019) and Shuttle Radar Topography Mission (SRTM) (Farr et al, 2007).
**3.1 Data Classification**
This study builds on existing multi-temporal debris flow datasets produced by Fan et al., (2019a). Dataset 1 has
an aerial extent of 892 km$^2$ and presents the location and dimensions of debris flow events between 2008 and
2020. Dataset 2 presents a list of mitigative actions e.g., construction of check dams, taken between 2008–2011.
We used an SRTM DEM to construct elevation profiles of Cutou, Chediguan and Xiaojia gullies to extract
topographic characteristics to understand the mechanism of slope failure in the event of a rainfall-induced debris
flow. These profiles facilitate morphological valley changes from debris flows to be identified. Through a
comparative analysis of the 20$^{th}$ August 2019 debris flows in Cutou and Chediguan, we investigated the relative
difference in land use change in the two gullies from 2008 to 2019, with a focus on changes before and after the
2019 flow event.
Landscape modification from 2005 to 2019 were mapped using high resolution (0.5 to 2.5m) satellite images
(Table 1). We selected images with less that than 50% cloud cover and cross-referenced the mapped features with
existing data sources in OpenStreetMap (OpenStreetMap contributors., 2023) and Dynamic World (Brown et al.,
2022). Where satellite imagery was unavailable, we used aerial photos obtained from Google Earth,
OpenStreetMap and World Settlement Footprint (World Settlement Footprint., 2019). It should be acknowledged
that platforms like OpenStreetMap offer a regional view of Wenchuan rather than a detailed local-scale with

mapping limited to main roads and 150 settlement polygons. However, this study's locations are unaffected by this due to their position next to the G213 national highway and G4217 road.

**Table 1** Satellite and aerial imagery used for data analysis and interpretation of the built environment.

| Data ID | Data Source | Acquisition Date | Resolution (m) |
|---|---|---|---|
| Aerial Satellite | Worldview (in QGIS – 'Satellite' XYZ tile) | 2022 | 1.0 |
| Satellite | Worldview (in Google Earth Pro., 2023) | 10.12.2010 26.04.2011 03.04.2018 29.10.2019 | 1.0 |
| Satellite | Planet | 14.08.2019 24.08.2019 | 3.0 |
| Satellite | Maxar Technologies (in Google Earth Pro., 2023) | 09.09.2005 26.04.2011 | 3.0 |
| Satellite | CNES/Airbus (in Google Earth Pro., 2023) | 15.04.2015 | 1.0 |

We used imagery collated from the sources listed in Table 1 to map manufactured features—including buildings, factories, roads, and dams—in order to understand the evolution of the built environment and subsequent human activities since 2005 across Cutou, Chediguan, and Xiaojia. We mapped features corresponding to human activities such as roads and properties. We highlighted at-risk zones in Cutou, Chediguan, and Xiaojia. We focus on spotlighting areas of high debris flow exposure in Cutou and Chediguan, comparing them with Xiaojia to evaluate the efficacy of check dams in mitigating potential debris flow hazards downstream of the dams.

**3.2 Modelling Future Debris Flow Runout and Building Exposure**

Both Cutou and Chediguan had check dams installed after the 2008 earthquake, while the Xiaojia gully remained unmodified. We compared the impacts of 2019 debris flows in Cutou and Chediguan gullies with a 2011 debris flow event in Xiaojia to identify the effectiveness of artificial dams in mitigating exposure to post-seismic debris flows. By using scenario modelling we identified at which point does the size of the hazard outweigh the mitigative capacity of the check dam to prevent overtopping. We mapped debris flows of differing scales within each of our three catchments using LAHARZ. LAHARZ is a GIS toolkit for lahar hazard mapping and modelling, developed by the USGS to calculate the area of inundation and cross sections based on empirical scaling relationships between area and volume (Schilling., 2014; Iverson et al., 1998). These empirical relationships allow for the creation of realistic inundation areas without a priory knowledge of the rheological parameters. The model simulates a debris flow triggered at a source point located on a digital elevation model and with an initial source volume. The model calculates the flow path downslope of the triggering location then generates a cross-section at each point downslope that represents the depositional volume for that area (Iverson et al., 1998).

We implemented this model using the extension in ArcGIS (Schilling., 2014). We used the 30m resolution DEM as an input, as it is the most reliable of the globally available DEMs. We identified the source areas of 2019 debris flows for Chediguan and Cutou and the 2011 for Xiaojia (Cutou – 351603, 3473449; Chediguan – 350846, 3453894; Xiaojia – 356666, 3439268) from satellite imagery and used these as the triggering locations for our simulations. We then prescribed three input volumes at each of these locations ($10^4 \ m^3, 10^5 \ m^3$ and $10^6 \ m^3$), The flow volumes simulate a range of observed post-2008 debris flows, representing low, high, and extreme debris flows documented in the Fan et al., (2019a) datasets. The volumes we selected reflects the range of similar hazard events in comparable geomorphological settings such as other parts of China and Italy (Wu et al., 2016; Bernard et al., 2019). For catchments with check dams, we added barriers at each check dam location by raising the cell count of the DEM by the height of the check dam obtained from field imagery.

The model was validated by comparing simulated runout extents with observed debris flows from post-2008 events. While a 30m resolution was the only available DEM for our study locations, we tested the sensitivity of DEM resolution on the extent of the final flow. A higher, 10m resolution DEM was available for the Cutou gully and we ran LAHARZ for that catchment. While the 10m DEM created a more effective flow path compared to the mapped data, the flow depositional area was similar in both the 10m and 30m scenario (RMSE 18m). Given the lack of a significant difference between the two DEM resolution we ran 30m scenarios across the three catchments. We note that there is not a strong understanding currently of what controls the maximum size of debris flows within Wenchuan catchments, hence we cannot attribute a particular probability to each scenario.

In the analysis of post-seismic debris flow, exposure and vulnerability assessments plays a crucial role (Lo et al., 2012). However, adapting traditional vulnerability methods which analyse inherent fragility and the potential loss of elements at risk, both attainable through remote practises, calculating exposure with minimal onsite data, remains a challenge. We adapted a vulnerability model by Zou et al. (2019) to quantify the extent of exposure to the built environment at our three sites, Cutou, Chediguan and Xiaojia.

Utilising satellite and/or aerial imagery and extracting spatial characteristics to identify both elements at risk as well as hazard-affected zones, our analysis facilitates the assessment of regional exposure without relying heavily on data collected onsite. All analysis steps were conducted within a GIS environment. Our model quantifies the susceptibility of the built environment to debris flow damage. The degree of exposure, $E_{df}$, is expressed as:

$$E_{df} = E_b \times C \pm M \qquad (1)$$

$E_b$ is the number of buildings damaged, and C is the fragility index of the elements at risk (Zou et al., 2019). Fragility values range from 0 to +1, with higher values indicating greater susceptibility to damage and/or failure. The assessment was conducted at the individual building level within GIS: building footprints were manually digitised and assigned fragility values based on proximity to debris flow channels and observed damage in previous events alongside literature. Due to limitations in detailed structural data and the reliance on remotely sensed satellite images, we simplified fragility to a binary classification: buildings clearly inundated, damaged or situated in highly susceptible locations (i.e., along the channel or gully mouth) were given a value of 1, all other buildings were set a value of 0. This approach provided us with a robust and replicable framework, avoiding overinterpretation of uncertain data. These values were then validated using historical damage reports, where available, from the 2008 earthquake recovery period to ensure applicability (Zeng et al., 2015; Wei et al., 2021; Petley et al., 2023). This GIS-based approach enables a replicable framework for similar hazard-prone contexts.

The key difference between our method and that of Zou et al (2019) is the incorporation a modification factor, M, to account for the effectiveness of engineered measures like check dams in mitigating building damage and subsequent exposure. The mitigation factor, $M$, quantifies the influence of engineered measures, in this study check dams, on the vulnerability and subsequent exposure of buildings to debris flow impacts. The addition of this factor brings an evaluative element to the exposure assessment, quantifying the influence of check dams and assigning values ranging from -1.0 to +2.0 to reflect a spectrum of mitigation outcomes:

- $M$ = **-1**: Effective mitigation of debris flows, resulting in a significant reduction in hazard exposure, as evidenced by a decrease in the number of buildings damaged during historical events following construction.
- $M$ = **0**: No mitigation present; exposure levels are entirely dependent on natural site conditions.
- $M$ = **+1**: Ineffective mitigation; there is no reduction in the number of buildings impacted in recorded debris flow events following dam construction.
- $M$ = **+2**: Mitigation increases exposure. Recorded events of similar volume show an increase in the number of buildings impacted following dam construction.

The above -1 to +2 scale was selected to capture a nuanced relationship between mitigation effectiveness and vulnerability. A reduction in $M$ (e.g., -1) lowers hazard exposure by reducing flow impacts at critical locations, thereby decreasing $E_{df}$. Conversely, an increase in $M$ (e.g., +2) elevates exposure, as development in hazard-prone areas amplifies the potential for damage. For example, a decrease in $M$ by one unit (from 0 to -1) reflects an improvement in flow attenuation due to effective check dams, reducing overall exposure. Conversely, an increase in M by one unit (from 0 to +1) signifies a scenario where mitigation fails, e.g. the 2019 debris flow event in Cutou, maintaining high exposure levels. At $M$ = +2, exposure exceeds natural vulnerability due to increased hazard presence caused by intensified land use near mitigation structures.

This scale was developed through a combination of evaluating present hazard mitigation and analysing of historical data, particularly from the 2008 earthquake recovery. Moreover, this approach, based upon the methodology proposed by Zou et al. (2019), allows for an assessment of exposure by considering both the physical resistance of buildings and the efficacy of mitigation efforts.

**4. Results**

**4.1 Assumptions**

In constructing the building inventory for Cutou and Chediguan, a comprehensive approach was taken to ensure accuracy and completeness. We used aerial and satellite imagery spanning 14 years, with a focus on mapping changes from 2011 to 2019. This involved careful analysis to delineate individual buildings, considering variations in size, shape, and spatial arrangement. Mapping efforts for Xiaojia were limited to 2010-2011 due to suboptimal image quality. Our approach incorporated assumptions regarding structural categorisation, including residential, industrial, and commercial buildings. These assumptions were informed by existing literature on local building typologies and architectural styles (Hao et al., 2013; Hao et al., 2012) and aerial photograph analysis from platforms such as Google Earth and Dynamic World. By amalgamating diverse information sources, we aimed to create a comprehensive inventory that correctly reflects the built environment of the study area.

Additionally, we used a 30-meter Digital Elevation Model (DEM) obtained from the SRTM dataset (Farr et al., 2007). However, it is necessary to acknowledge the limitations of this data, particularly its low resolution and subsequent blockiness, which potentially hindered detailed topographical analysis. Despite this, the DEM provided valuable contextual information for understanding the terrain and its influence on building distribution and spatial patterns within the three sites. Furthermore, while using the empirical LAHARZ model for debris flow inundation mapping, we had to account for a degree of approximation in both aerial coverage and debris flow inundation due to the 30m resolution of the DEM file.

**4.2 Mapping Post-Earthquake Risk**

Analysis of satellite imagery from 2005 to 2019, and topographic profiles, reveals channel widening, deepening, aggradation, and deposition, likely attributed to the mobilisation of coseismic deposits and subsequent debris flow occurrences (Zhang et al., 2015; Wang et al., 2018) (Fig 3). These observations allowed us to determine the zones of erosion, transportation, and deposition for each gully and to track changes over time. Hydrological and geomorphological analysis examines landscape morphology to identify erosional and depositional features i.e., scarring, changes to river channel, sediment buildup (Fig 4). By integrating the above, we delineated erosion-prone areas, which permitted sediment transport routes to be approximated, and identify locations of sediment deposition along the hydrological profile.

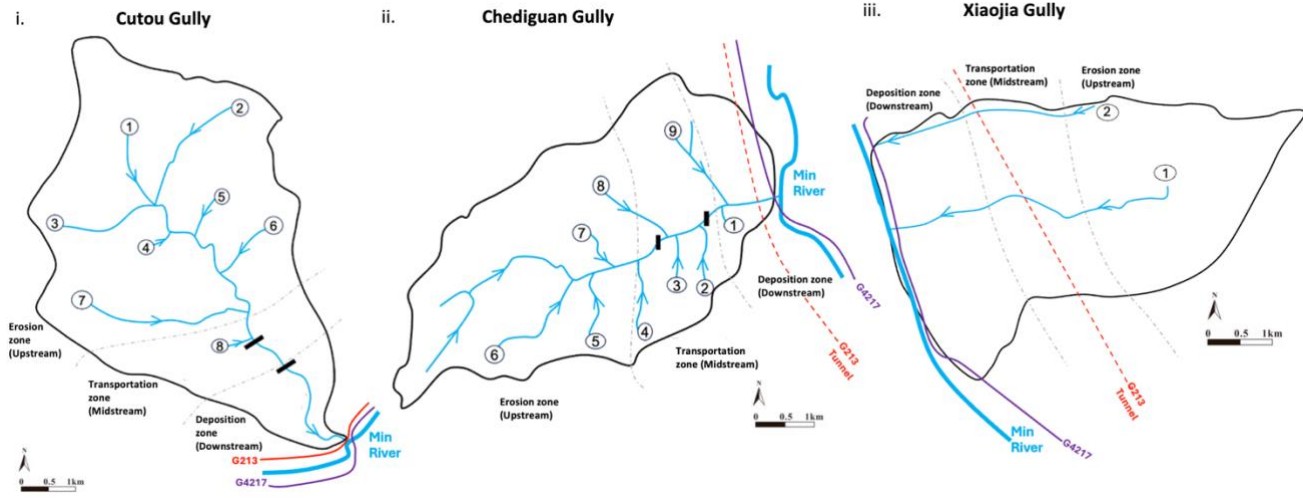

**Figure 3:** Hydrological profiles for the 3 study sites. Dam locations approximated for Cutou (i) and Chediguan (ii) based on a combination satellite imagery. Streams and main tributaries are numbered to identify and reference key branches within each catchment. Catchment profiles have been segmented into 3 zones – 'erosion', 'transportation' and 'deposition' and key infrastructure annotated.

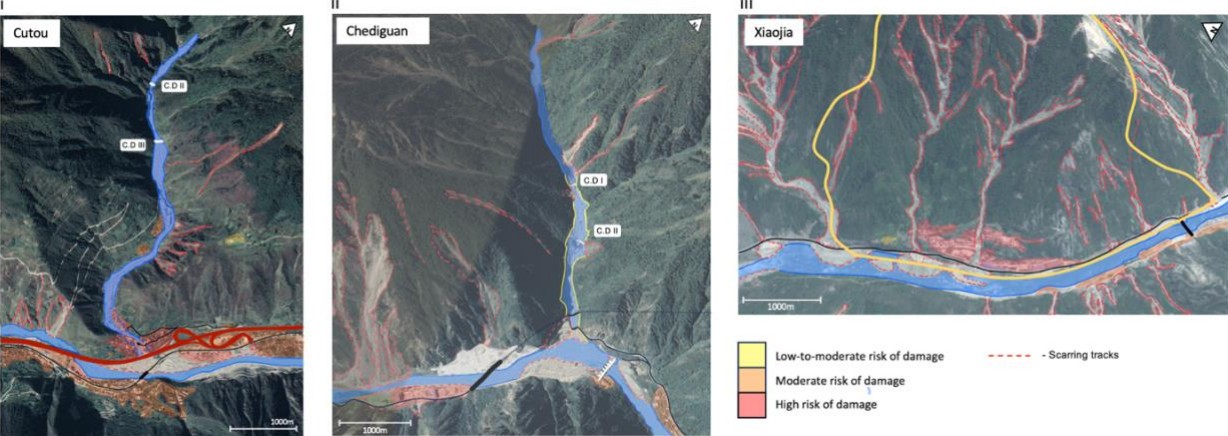

**Figure 4:** Satellite images of the 3 study locations highlighting areas of scarring from previous debris flow activity
and areas of increased erosion (© Google Earth 2019)**.** Dam locations have been approximated for Cutou (i) and
Chediguan (ii). The built environment has been shaded based upon risk of damage based upon proximity to areas
of high erosion. Critical infrastructure has been added where appropriate: black like represents the G4217 road
with the thicker sections representing bridges and the dashed lines, tunnels. The red line in (i) is the G213
Highway.
In Cutou and Chediguan, deposition patterns shifted post-earthquake, particularly following the construction of
check dams. Increased deposition occurs behind check dams compared to meander bends and basal slopes of the
debris fan, demonstrating the effective sediment trapping of the check dams (Wang et al., 2019). Regarding the
erosion patterns in Xiaojia, we observed common patterns in the upper gully sections at higher elevations, with
deposition occurring at the basal slopes. This is due to the absence of structural alterations to the channel,
permitting sediment to be transported to the channel and subsequent river outlet directly. The deposition patterns
in Cutou and Chediguan, are strongly controlled by the distribution of check dams, in the middle and downstream
portions of the catchment (Wang et al., 2019). The complex interplay between natural and anthropogenic factors
demonstrates the dynamic evolution of risk in post-earthquake catchments and highlights the role of check dams
in both mitigating and potentially exacerbating risk.
The landscape morphology prior to the 2008 earthquake was marked by extensive vegetation (over 70% of land
cover) and minimal permanent engineered features. Cutou gully contained a widespread distribution of buildings
along both the mid and lower slopes. Figure 5 shows the growth of the built environment between 2005 and 2019
in Cutou and Chediguan and between 2010 to 2011 in Xiaojia. The built environment in Cutou is concentrated
within the transportation and deposition zones on both sides on the stream. In Chediguan by comparison we
observed fewer residential structures, mostly industry and some commercial structures. Additionally, buildings in
the gully are more spread out than in Cutou highlighted by the isolated settlements to the south of the catchment
and single industrial site situated in the basin. Post-2008, noticeable tracks of scarring from debris flows are
concentrated downstream of dams 2 and 3 in Cutou (Fig 4(i)), and upstream of dams 1 and 2 in Chediguan (Fig
4(ii)). Deposition patterns are evident downstream of all modifications, forming a depositional zone,
encompassing approximately 15% and 20% of the built environment in 2019 within the transportation zone of
Cutou and Chediguan, respectively.
Xiaojia was chosen as the comparative catchment due to the absence of engineered mitigation such as check dams.
This analysis of Xiaojia therefore enables comparisons on the effectiveness and limitations of engineering
approaches applied to Cutou and Chediguan. In Xiaojia, the lack of engineered dam structures, results in different
erosion and deposition patterns compared to the other two catchments. Distinct patterns of upstream erosion and
downstream deposition are observed, contrasting with the more controlled environments in the modified gullies,
where deposition occurs on the northern channel flank and pronounced erosion on the southern flank. The data
availability for building types, quality and spatial distribution was limited to remote sensing images and few
literature sources, which restricts our ability to thoroughly assess how specific building characteristics, such as
materials, influence the exposure of the built environment to debris flow hazard. This is particularly evident in
Xiaojia, where more specific input data would be beneficial for understanding the role of urbanisation and
construction practices on risk levels.

Our analysis of Xiaojia unveils no discernible relationship between building development and heightened exposure, particularly to residential and critical infrastructure. This lack of correlation is potentially linked to factors beyond simple urbanisation patterns, like construction quality, building regulations, presence of natural barriers, and effectiveness of mitigation measures. Natural terrain barriers observed in this gully including steep slopes and rocky outcrops, could limit the extent of debris flow impacts by reducing the mobility of debris and offering natural protection to certain areas. To fully understand this observation, further investigation into the above variables is warranted. The absence of significant urban expansion, particularly post-earthquake in Xiaojia may be a key factor in mitigating exposure. This area has experienced less intensive development compared to Cutou and Chediguan, where urban expansion following the implementation of check dams potentially increased exposure to debris flow hazards. Furthermore, the building quality and structural characteristics in Xiaojia may play a significant role in influencing its overall vulnerability and subsequent damage outcomes. Due to limited detailed building-specific data in terms of construction and materials, our assessment simplifies vulnerability to a binary classicisation based on observed damage and location. It is possible that buildings in Xiaojia may be of higher structural integrity or designed to withstand environmental stressors better than those in more developed catchments which would contribute to the observed exposure patterns.

Additionally, detailed mapping of past debris flow events and their impacts on the built environment could provide insights into the specific mechanisms influencing vulnerability in Xiaojia. By conducting a more comprehensive analysis that considers these factors – especially in terms of land-use planning, construction standards and the role of natural terrain features at the local scale, we can gain a better understanding of the complex interactions between building development and exposure to natural hazards in Xiaojia. This, in turn, can inform more effective risk management and mitigation strategies tailored to the unique characteristics of the area. Development in Xiaojia primarily concentrates on the lower slopes (Fig 5(i) and (ii)) at the gully mouth, featuring the construction of major roads and highways (G213 and G2417), alongside the expansion of existing residential areas. Chediguan exhibits a less marked land cover transformation, owing to roads being directed through mountain tunnels. Notably, development in Xiaojia mainly surges post-earthquake up to 2010, with only minor construction activities documented thereafter (Fig 5(iii); see also Supplementary Figures S3(i) and 3(ii))

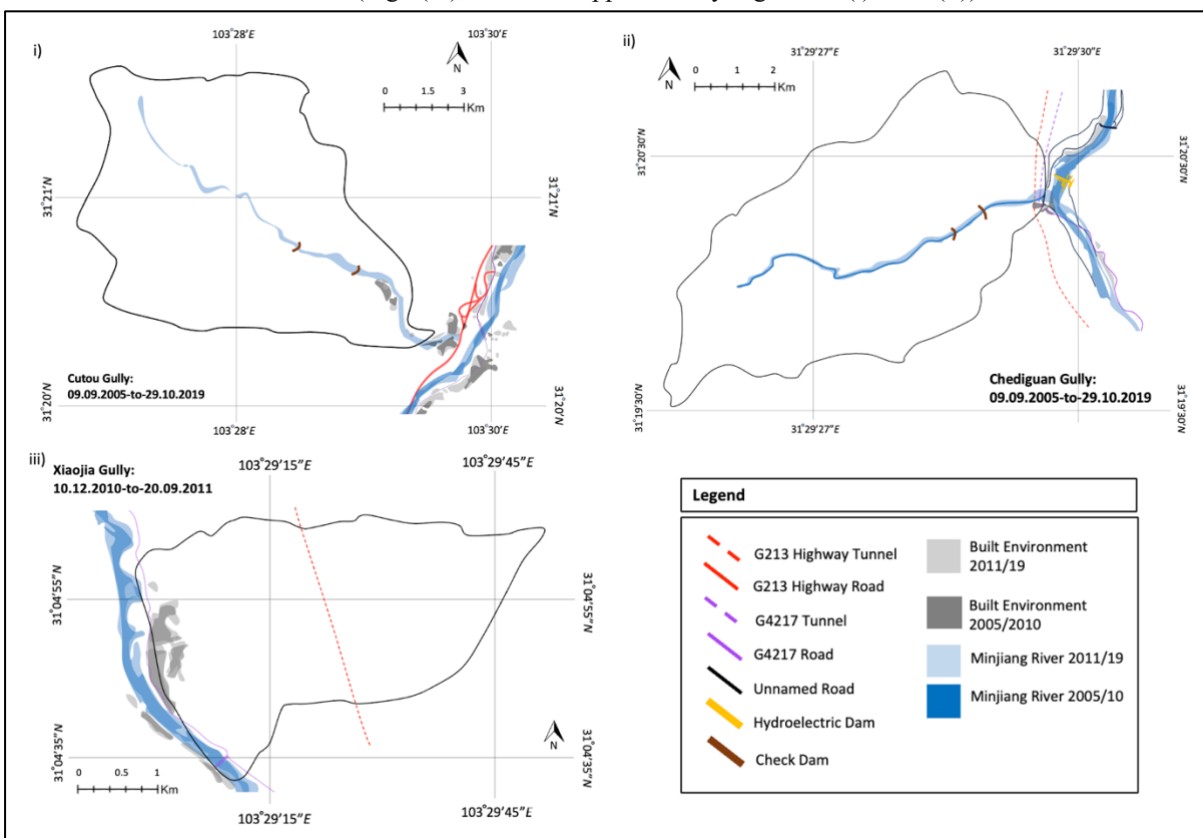

**Figure 5:** Evolution of the built environment and key infrastructure in (i) Cutou, (ii) Chediguan and (iii) Xiaojia post-earthquake between 2005 and 2019. Roads and tributary channels are annotated; all symbology is defined in the legend. In Cutou, pale blue shading indicates the mapped extent of a tributary channel post-2010 and does not imply absence in the earlier period. Several areas shown as built environment in 2005–2010 may have been damaged during debris flows, but limitations in imagery restrict differentiation between reconstruction and new

development in subsequent years. Scale bars have been corrected to reflect the true spatial extent of each catchment.

We mapped the number of buildings impacted by debris flows that occurred within the Chediguan and Cutou gullies. At 02:00 a large-scale debris flows hit Chediguan, impacting numerous structures, at around 05:00 a similar debris flow hit Cutou with significant inundation noted. 79 of the 197 buildings (40%) in Cutou (Fig 5(i) and Supplementary Figures S1(i) and (ii)) were impacted by the flow i.e., flooded, damaged, or destroyed. Buildings in Chediguan were less impacted by that event with 7 out of the total 69 (10.1%) (see Supplementary Figures S2(i) and (ii)). We combined the satellite imagery with the datasets produced by Wang (2022) which supported our observations of check dam overtopping in both Cutou and Chediguan during the 2019 event. In 2011 a similar event in Xiaojia impacted approximately 5 of the 43 (11.6%) buildings in the gully (Fig 5(iii)).

## 4.3 Modelling exposure to post-earthquake debris flows

Our LAHARZ simulations demonstrate a clear correlation between exposure and debris flow runout, revealing a notable increase in building damage as runout volumes escalate from low (10,000m$^3$) to high (100,000m$^3$) and extreme (1,000,000m$^3$) scenarios across all catchments. Despite the presence of check dams, the 2019 debris flows recorded runout volumes significantly larger than the maximum simulated volume, resulting in substantial building and infrastructural loss in Cutou (Fig 6(i)) and Chediguan (Fig 6(ii)).

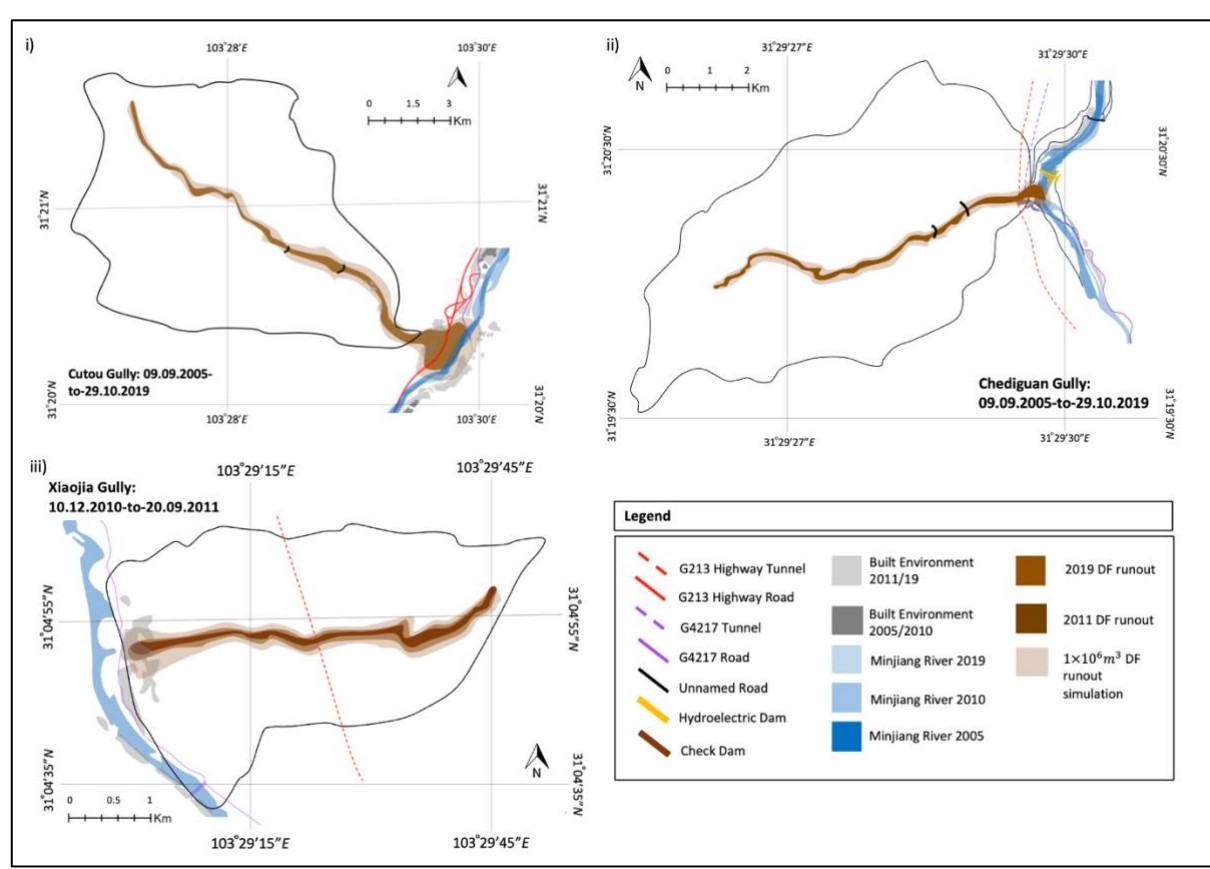

**Figure 6:** Debris flow runouts for 2019 in Cutou (i) and Chediguan(ii) and 2011 in Xiaojia (iii) underlain by the extreme LAHARZ runout scenario. Low (10,000m$^3$) and high (100,000m$^3$) runouts are not displayed as they are not easy to visualise at map scale.

While low (10,000m$^3$) and high (100,000m$^3$) volume runouts are not visually represented in figure 6 dues to their small spatial extent that do not clearly illustrate the influence of check dams – our analysis highlights their critical role at the smaller scale. These smaller simulations demonstrate that check dams effectively reduce exposure during low magnitude debris flow events, limiting damage to building and infrastructure in Cutou and Chediguan. Thus, although not easily visualised at the map scale, the efficacy of check dams in mitigating small debris flow events is not disputed by our results.

We examined the temporal dynamics of building changes within the three gullies in response to check dam development, while also considering the implications of the levee effect (fig 6). Our simulations revealed the

effectiveness of engineered measures in mitigating exposure to debris flow events. In both Cutou and Chediguan, the presence of check dams led to reduced exposure at low and high debris flow volumes (fig 7(i) and (ii)). However, the mitigative structure provides no discernible protection against extreme debris flows. Notably, Cutou consistently exhibited elevated exposure to debris flow runout compared to Chediguan. In contrast, the unengineered Xiaojia (fig 6(iii)), shows a more consistent increase in exposure with debris flow volume, illustrating the effectiveness of check dams at low and high debris flow volumes. This comparison underscores how unmitigated gullies, like Xiaojia, experience greater susceptibility to debris flow damage compared to engineered gullies like Cutou and Chediguan at the lower to moderate volume. It is important to clarify that Figure 6(iii) depicts the cumulative built environment up to 2019 in Xiaojia, including all infrastructure developed post-2005. When we refer to "restrained expansion" in Xiaojia between 2011 and 2019, this relates specifically to the comparatively limited rate and spatial extent of new built environment development during that period, as shown in Figure 5.Xiaojia's post-2011 expansion appeared restrained, indicating a potential adaptive response following debris flow events. In contrast, substantial expansion occurred in Cutou and Chediguan between 2011 and 2019, despite experiencing a debris flow event in 2013, suggesting the impact of check dams implemented post-2013.

Furthermore, the incremental increase between high and extreme simulations in Xiaojia paralleled Chediguan's gradual incline, diverging from Cutou's steep escalation. Xiaojia sustained a maximum building damage of 33% under extreme scenarios, compared with 59% in Cutou and 22% in Chediguan. This discrepancy suggests that the effectiveness of check dams may have limits under extreme debris flow events, highlighting that while check dams may reduce damage at low to moderate volumes, they provide limited protection during extreme events. Our observations underscore the nuanced variability in the effectiveness of check dams, influenced by contextual factors and landscape characteristics.

However, it is important to note that our study only explores this hypothesis within a small sample, three catchments, which contains the ability to generalise these findings. While the evidence points to a potential levee effect associated with check dam construction and subsequent bult environment expansion, further research across a larger catchment sample is necessary. Expanding the scope of this analysis would help validate whether the trends we have observed hold more broadly and improve understanding of socio-environmental feedback on risk exposure and mitigation effectiveness.

480

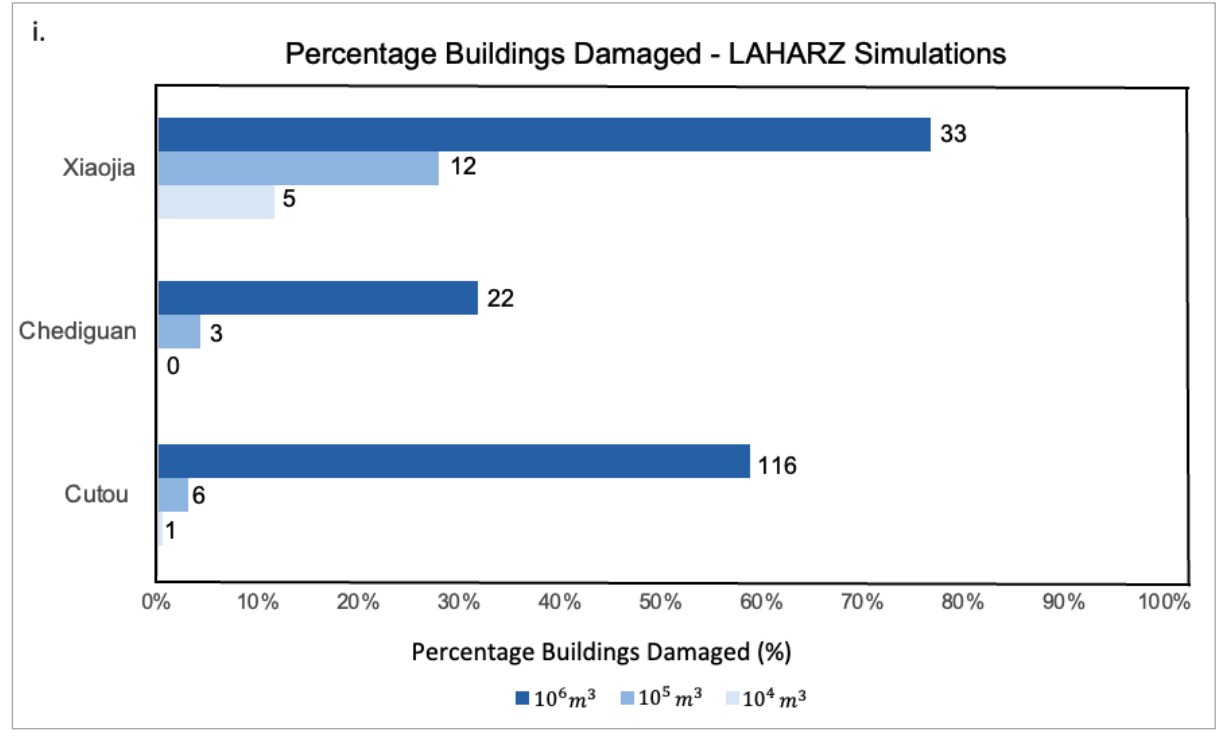

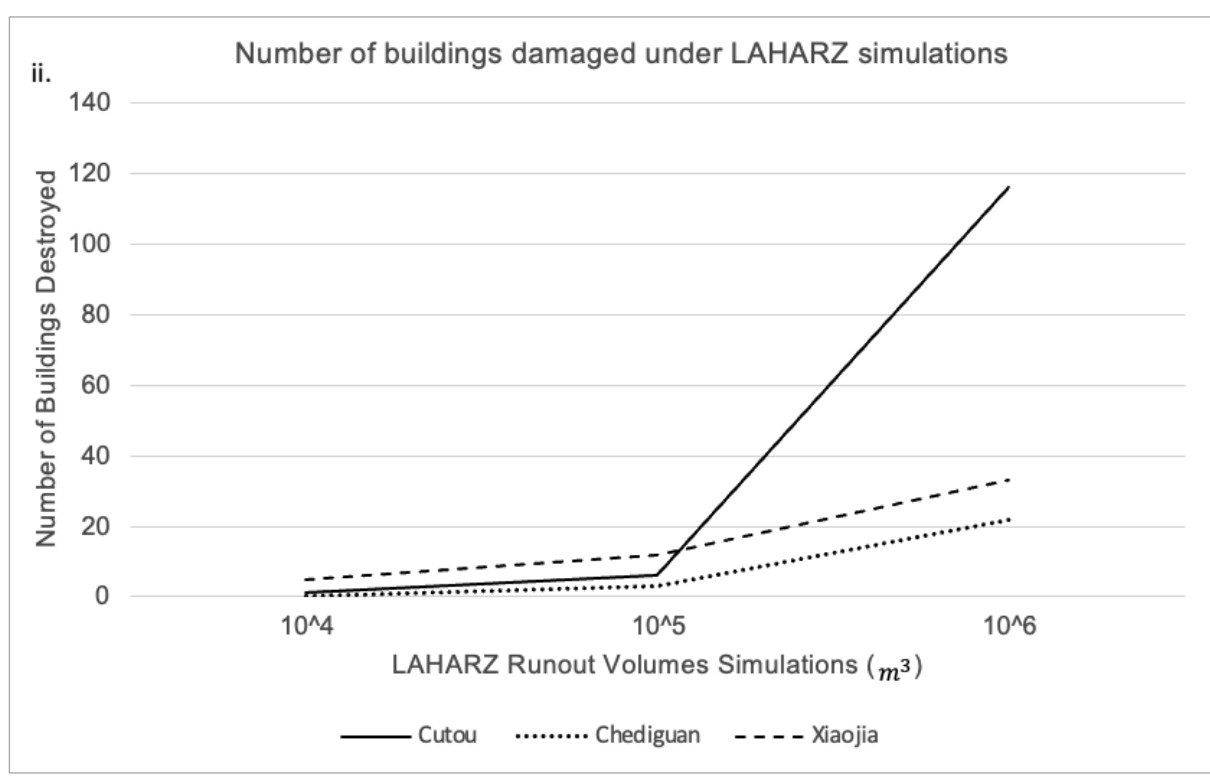

**Figure 7:** Built environment impacts from three debris flow scenarios modelled using LAHARZ at Cutou, Chediguan and Xiaojia. (i). Percentage of buildings damaged as a proportion of total buildings (Cutou – 197, Chediguan – 69 and Xiaojia – 43) in each scenario. The numbers at the end of the bars are the number of buildings damaged buildings in that debris flow scenario. (ii) Total number of buildings damaged by each simulated debris flow.

Figure 7 illustrates how a tenfold increase in runout volume corresponds to building damage, with a discernible rise in impacted building numbers noted between low and high scenarios, and a significant incline between high

and extreme scenarios across all catchments. The significant jump in destruction observed in Cutou between 10^5
and 10^6 debris flow volume simulations is driven by the combined effect of the increased flow magnitude
overwhelming the capacity of check dams and the spatial distribution of building within the flow path, leading to
disproportionately higher damage. These simulations provide valuable insights into the efficacy of engineered
mitigation structures. While check dams in Cutou and Chediguan effectively reduce exposure at low and high
runout volumes, concerns arise when surpassing the maximum capacity.
We acknowledge the importance of decoupling debris flow inundation from building damage, as damage depends
on various factors including but not limited to building materials and structural integrity, which are not controlled
for in this analysis. Therefore, our assessment focuses primarily on exposure as a proxy for risk. Regarding the
influence of check dams on sediment dynamics, these structures alter the distribution of erosion and deposition
by trapping debris upstream (Figure 3), thereby reducing downstream sediment loads and runout distances. This
function contributes to reduced exposure in low to moderate debris flow scenarios but become less effective during
extreme events when sediment volume exceeds retention capacity.
Urbanisation emerges as a significant contributing factor impacting exposure and future risk, with the presence of
check dams during the 2019 events significantly contributing to the built environment's exposure. However, due
to the lack of available data on building materials in these three regions, we were unable to quantify their influence
on structural vulnerability. To fully understand the effect of check dams and validate our statistical approach,
comprehensive numerical analysis of multiple hazard events in each gully is necessary. This sub-section addresses
the elements driving hazard-related risk scenarios, including the trigger event, return period, and level of damage,
and underscores the importance of considering these factors when suggesting and implementing modifications.
The exposure model is applied to historical events (2019 and 2011) and LAHARZ simulations, showcasing
changes in the degree of exposure across the catchments with increasing debris flow runout volumes (Fig 8).
Consistent with earlier observations in exposure, Cutou exhibits a heightened vulnerability to debris flows at 64%
after the 2019 event, followed by Chediguan at 52% and Xiaojia with 2% in 2011. A discernible change in building
exposure is observed between the high and extreme scenarios across all catchments. The most influential factor
in overall vulnerability remains the number of buildings, highlighting urbanization as a contributing factor
impacting both exposure and physical vulnerability. Moreover, the presence of failed check dams in Cutou and
Chediguan during the 2019 events significantly contributes to their physical vulnerability. These failures occurred
primarily through overtopping of the dams, which exacerbated the impact of debris flows in these catchments.

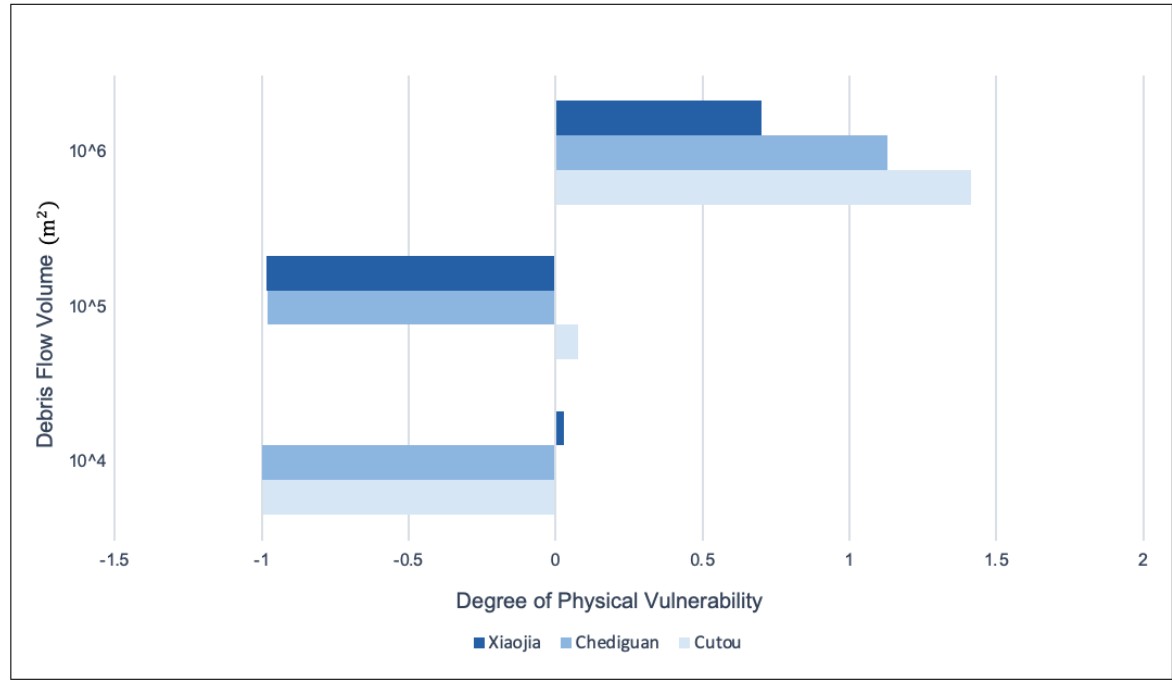

**Figure 8:** Changes in the degree of exposure with increasing runout volumes using the exposure model developed
in equation 1. The 2011 and 2019 debris flows are also noted as a base marker from an observed hazard event.

## 5. Discussion

Post-earthquake structural interventions influence the volume and spatial distribution of sediment within the catchment. Our observations show that check dams act as local depocentres within the catchment, often storing large volumes of sediment upstream of the majority of building development. The choices made about post-earthquake development of the built environment, particularly housing, and mitigative measures like check dams, evolve rapidly without a clear approach to mitigating adverse long-term consequences of sediment retention behind dams (McGuire et al., 2017). Additionally, the processes driving geological disasters in the complex landscape of the Longmenshan occur at different timescales to the rapid socio-economic development in the region (Chen et al., 2022).

Although our analysis focuses on smaller-scale communities, the implications drawn from our findings echo those of broader studies. For instance, Arrogante-Funes et al. (2021)'s extensive investigation into hazard mitigation strategies in larger geographical regions, drew parallels to the effectiveness and limitations of mitigation measures to debris flows. Similarly, Chen et al. (2021) provided insights into the complexities of hazard mitigation, emphasising the necessity of adaptive responses considering local contexts. This aligns with our analysis that each gully must be assessed and mitigated individually rather than collectively to account for local geological and hydrological influences on mitigation effectiveness. Moreover, Li et al. (2018) examined the long-term impact of engineering interventions, noting the variability in check dams effectiveness over time. This supports our conclusion that the diminishing effectiveness of check dams is likely the result of sediment accumulation and structural degradation and highlights the necessity for their continued maintenance post-construction in addition to adaptive mitigation strategies. Furthermore, Eidsvig et al. (2014) and Tang et al. (2011) explored the interplay between socio-economic factors and hazard vulnerability, emphasising that community resilience is directly linked to economic resource availability and social cohesion. This corroborates our understanding that debris flow mitigation is a multifaceted issue, and socio-economic conditions are integral to their success. By situating our findings within the broader context delineated by these studies, we accentuate the relevance and applicability of our research beyond the confines of the specific communities under study.

Open check dams, similar to those established in Cutou and Chediguan, play a pivotal role in bed stabilization, slope reduction, and the regulation of sediment transport (Bernard et al., 2019). However, inadequate understanding of post-earthquake debris flow characteristics has led to the failure of many newly constructed engineered structures to mitigate hazards effectively, amplifying damage instead (Chang et al., 2022). During the August 2019 debris flow, Cutou experienced the highest inundation, with 40% of surveyed structures directly affected, including critical infrastructure like the G4217 highway bridge. In Chediguan, despite a declined industrial presence, compared to earlier periods i.e., during the construction of the hydroelectrical dam in the Minjiang, debris flow impacts affected 7% of structures. The presence of check dams in both locations contributed to raised exposure and hazard impacts during the 2019 event, with overtopping and damage to dam sections recorded.

We conducted LAHARZ scenarios to predict potential exposure to debris flows with volumes that have been observed within the catchments and the region. While it is intuitive that larger debris flows would affect a greater areas, our results provide a quantified and site-specific correlation between exposure and debris flow runout, showing notable increases in building damage as runout volumes escalated from low to extreme across all catchments. This explicit demonstration is critical for understanding the scale of risk in these environments. We observed two key elements to the role of check dams in affecting exposure to debris flows. When empty, check dams are effective at mitigating the effects of small and medium volume debris flows. Yet, the check dams in Cutou and Chediguan, are from our results, designed to mitigate small to medium debris flow volumes, which limits their effectiveness against the extreme runout volumes denoted by our simulations. In other words, it is not that check dams are inherently ineffective against large flows, but within the specific context of the mountainous Sichuan landscape, they are insufficient to fully mitigate these extreme debris flow events. Additional simulations without check dams at Cutou and Chediguan indicated that while check dams did reduce damage from smaller events as shown in Figure 7, their failure during extreme events from overtopping or breaching can exacerbate impacts releasing stored sediment, sometimes resulting in greater damage than scenarios without dams along sections of the gully channel. Large runout volumes in the 2019 debris flows resulted in substantial building and infrastructural loss in both Cutou and Chediguan, suggesting a negative contribution from damaged check dams. Cutou was found to be highly exposed to extreme debris flow volumes, a result of its high degree of urban development concentrated at the basal slopes.

The fact that Xiaojia was found to possess the least exposure to the most extreme debris flow volume suggests that there may be an adaptive component to debris flow mitigation in catchments without significant check dam

development. These findings suggest that urban development and debris flow risk co-evolve based on the nature
of the structural interventions the studied areas.
Our analysis of erosion, transportation, and deposition zones for each gully revealed significant changes in
landscape morphology post-earthquake, likely attributed to mobilised coseismic deposits and subsequent debris
flow occurrences. The presence of check dams influenced deposition patterns, with mid-to-downstream trends
indicating effective sediment retention in Cutou and Chediguan, while Xiaojia exhibited typical erosion-
deposition behaviour. Our findings can be supported by a similar occurrence during the "8.13" debris flow event
in Wenjiagou. The damage and subsequent failure of mitigative check dams led to the inundation of 490 houses
or more recently, a debris flow in the Miansi and Weizhou townships on 27 June 2023 blocked the valley in the
first instance before breaching the dam and causing 7 fatalities (Petley., 2023). Further research is thus imperative
to devise appropriate mitigation approaches for post-seismic debris flows. Whilst existing literature has
underscored the physical effectiveness of check dams in reducing exposure to debris flow impacts within Alpine
terrains (Piton et al., 2016), it should be noted that their primary function extends beyond this to also provide
socio- economic and political reassurance (Wu et al., 2012; Chen et al., 2022)
The findings of our paper provide preliminary evidence supporting the theory of the levee effect by demonstrating
how the implementation of mitigative measures, such as check dams, can inadvertently increase exposure levels
and risk perception in hazard-prone areas. However, we acknowledge that our analysis is limited to three
catchments, and additional socio-economic and geographic factors may also encourage or discourage
development. Therefore, further research with a wider study sample is needed to fully substantiate. The interplay
between engineering solutions and the built environment as highlighted in our study through analysis of the 2011
and 2019 events as well as the LAHARZ simulations, illustrates the levee effect. Similar to previous studies on
flooding and the levee effect. Similar to previous studies on flooding, (e.g. Collenteur et al., 2015), our paper
suggests that the perceived reduction in hazard risk due to mitigative structures can lead to increased levels of
exposure due to raised development in debris flow-prone regions. This effect is particularly evident in the Cutou
catchment where urban expansion occurred post-dam construction, despite the repeated occurrence of high
magnitude debris flows. This suggests a distorted perception of hazard risk, which ultimately drives urbanisation
into vulnerable areas (Chen et al., 2015; Ao et al.,2020).
The levee effect can influence exposure to large-scale debris flow events by inadvertently increasing risk in areas
protected by engineered mitigation structures, such as check dams. This occurs because the perceived safety
provided by these structures can encourage development in vulnerable areas, which might otherwise remain
uninhabited due to their high-risk nature. This phenomenon is best evidenced in our paper by the Cutou catchment,
where the construction of check dams in 2013 coincided with widespread urban expansion, despite ongoing small-
scale debris flow activity in the area. It is also possible that the prioritisation of check dam construction was
influenced by pre-existing high levels of development, reflecting a reactive approach to hazard mitigation in
already urbanised zones. In our study, we have observed that the rate of urban expansion post-dam construction
increased. This raises an import ant question about the interplay between mitigation efforts and development
patterns, suggesting that structural interventions may both respond to and shape urban growth in hazard prone
areas. Subsequently, building exposure increased by 64% post-2008, underscoring the risk amplification
associated with structural mitigation. This observation highlights the necessity of coupling structural interventions
with strategies that address residual risks and foster community awareness of long-term hazard vulnerabilities.
The 2019 debris flow event exemplified the risks associated with this effect, as the flow overtopped the check
dams and used the stored material as a secondary fuel, significantly amplifying the impact. As a result, 40% of
surveyed buildings were inundated, demonstrating how the levee effect can potentially escalate exposure to large-
scale debris flow events.
Our LAHARZ simulations further reinforce the limitations of engineered structures as the sole mitigative measure
in alpine environments; urbanisation of mountainous terrains further complicates the balance between
technological advancements and geological hazards (Zhang, S et al., 2014; Zhang and Li., 2020; Luo et al., 2023).
Despite the presence of check dams, our extreme runout volume resulted in significant impacts on the built
environment in Cutou and Chediguan, including overtopping and dam failure. The use of these simulations
emphasises the challenges of reducing exposure to at-risk structures and highlights the unpredictable nature of
debris flow occurrences. Moreover, our findings relating to the altered patterns of erosion and deposition
emphasise the relationship between natural topography, engineered interventions, and risk perception in post-
seismic debris flows. Urbanisation exacerbates this complexity, influencing exposure and physical vulnerability
through deposit remobilisation. Our LAHARZ simulations serve as a practical demonstration of the levee effect,
illustrating how engineered structures may not provide adequate protection against runout volumes similar to the
extreme simulation, thereby reinforcing the importance of considering the levee effect in debris flow risk
management. The unpredictable nature of debris flow occurrences from pinpointing their location and timing to
ascertaining their volume and velocity ultimately means that the concept of the 'levee effect' remains core to the
issue of debris flows in post-seismic Sichuan (Cucchiaro et al., 2019a; Tang et al., 2022).
Whilst our findings are not able to definitively determine the prevalence of the levee effect with regards to
development in post-seismic environments like Sichuan, we hypothesise that the implementation of mitigative
structures like check dams may inadvertently increase exposure levels to large-scale debris flow events by creating
a false sense of safety. Although our investigation does not fully explore this phenomenon, our outcomes suggest
that the development of infrastructure in areas perceived to be safe due to the presence of engineered structures
may amplify hazard exposure. This highlights the limitations of solely relying on engineered interventions in
reducing exposure to at-risk structures under the extreme LAHARZ scenario. Furthermore, we highlighted the
complex interplay between engineering solutions and human behaviour, warranting further investigation
(Papathoma-Köhle et al., 2011; Gong et al., 2021). By emphasising the challenges and limitations of engineered
structures in mitigating debris flow impacts, we underscore the need for comprehensive risk management
strategies that consider the complexities of urbanization and flow-based hazards in mountainous terrains.
Despite the presence of these engineered interventions, our analysis demonstrates significant exposure levels and
infrastructure damage during extreme debris flow events. This discrepancy between perceived risk reduction and
actual hazard exposure underscores the need for a more comprehensive understanding of risk perception in the
context of hazard mitigation strategies. Moreover, our study highlights the importance of considering human
behaviour and decision-making processes in the design and implementation of risk management measures. Future
research should focus on elucidating the mechanisms driving risk perception in hazard-prone areas and developing
strategies to bridge the gap between perceived and actual risk to enhance the effectiveness of mitigation efforts.
**6. Conclusion**
Our study investigated the changing exposure to debris flows in Cutou Chediguan and Xiaojia since the 2008
Wenchuan earthquake. We used high resolution satellite imagery to build a time series of building inventories
between 2005 and 2019. Despite recurrent debris flow occurrences between 2010 to 2013, we observed increased
urban developments across all three gullies to varying extents until 2015.
We identified significant differences in the impacts of debris flow events in 2011 and 2019 respectively.  In the
August 2019 debris flow, Cutou experienced the highest inundation, with 40% of surveyed structures directly
affected, including critical infrastructure such as the G4217 highway bridge. In contrast, the 2011 event in Xiaojia
impacted approximately 11.6% of buildings in the gully, indicating a lower level of damage compared to Cutou.
The presence of check dams in Cutou and Chediguan contributed to increased exposure and hazard impacts during
the 2019 event, with overtopping and damage to dam sections recorded at both locations. However, despite the
presence of these mitigative structures, the impact on the built environment was significant, suggesting limitations
in their effectiveness, particularly during extreme runout volumes. Our Laharz simulations demonstrated a clear
correlation between exposure and debris flow runout, revealing a notable increase in building damage as runout
volumes increased from low to high and finally extreme scenarios across all catchments. Despite the presence of
check dams, the simulations indicated that these structures were unable to reduce the impacts on the built
environment, especially during extreme events. Furthermore, our analysis highlighted a heightened level of built
environment exposure in Cutou compared to Chediguan and Xiaojia driven by urbanisation, the presence of
critical infrastructure, and the effectiveness of mitigative measures.
Our findings suggest that the presence and location of check dam in gully channels likely increased building
exposure by fostering a perception of reduced hazard risk, thereby contributing to a levee effect. This raises
concerns about the long-term implications, including  structural integrity, maintenance and clearing. LAHARZ
modelling provides comprehension of check dam efficacy, raising concerns for Cutou and Chediguan in high-to-
extreme runout events. Further, the combined use of the LAHARZ GIS toolkit and exposure analysis contributes
to a holistic understanding of the risk landscape, informing strategies for enhanced disaster resilience and
sustainable development in vulnerable areas.
The assumptions and subsequent considerations highlighted throughout our study underscore the complexities of
how check dams, as a mitigative structure, influences land-use planning and development in hazard-prone areas.
These factors ensure that the data outputs are comprehensive but also reflective of the inherent complexities of
the study area and limitations in available data sources and analytical tools. We have highlighted a relationship
between the presence of engineered measures like check dams alongside the built environment, showing how this
relationship has contributed to increased debris flow impacts post-2008 earthquake in Sichuan, particularly

provinces along the Minjiang. Our results emphasise the need for a multi-faceted approach to risk management, integrating socio-economic development and addresses the paradoxical role of mitigative structures in shaping public perception to hazard exposure and vulnerability. Understanding these complexities is vital for informed decision-making and effective debris flow risk management.

Overall, our findings have indicated that the 2019 debris flow events caused more significant damage and higher exposure levels compared to the 2011 flow, emphasising the need for comprehensive risk management strategies in debris flow-prone areas.

**Acknowledgements**

EH is supported by the BGS-NERC National Capability grant 'Geosciences to tackle Global Environmental Challenges' (NERC reference NE/X006255/1) and publishes with permission from the Executive Director of the British Geological Survey.

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

## Statements & Declarations

### Conflict of Interest
The authors disclose no financial or non-financial interests of competing interest during the preparation of this manuscript.

### Author Contribution
All authors contributed to the study conception and design. Material preparation, data collection and analysis were performed by IU, TH and EH. The first draft of the manuscript was written by IU and all authors commented on previous versions of the manuscript. All authors read and approved the final manuscript