# Peer review of "Transformations in Exposure to Debris Flows in Post-Earthquake Sichuan, China"

_EGUsphere, 2024_

## Author Comment (AC1)

1. **Lines 143-226: The methodology section should include the rationale for the selection of model parameters and provide a sensitivity analysis to enhance the credibility of the model results.**

We have revised the model section of our methodology to include a detailed rationale for the selection of LAHARZ. This now explains the factors we considered when selecting each parameter including their relevance to the study area and impact on model accuracy. Furthermore, we have incorporated a sensitivity analysis of the DEM on the model's output. This addition enhances the robustness of the results and justify the model's applicability in the context of our research. Please see additions, in italics, to be added to the revised manuscript below:

*LAHARZ is a GIS toolkit for lahar hazard mapping and modelling, developed by the USGS to calculate the area of inundation and cross sections based on empirical scaling relationships between area and volume (Schilling., 2014; Iverson et al., 1998). These empirical relationships allow for the creation of realistic inundation areas without a priory knowledge of the rheological parameters. The model simulates a debris flow triggered at a source point located on a digital elevation model and with an initial source volume. The model calculates the flow path downslope of the triggering location then generates a cross-section at each point downslope that represents the depositional volume for that area (Iverson et al., 1998).*

*We implemented this model using the extension in ArcGIS (USGS., 2007). We used the 30m resolution ASTER DEM as an input, as it is the most reliable of the globally available DEMs. We identified the source areas of 2019 debris flows for Chediguan and Cutou and the 2011 for Xiaojia (Cutou – 351603, 3473449; Chediguan – 350846, 3453894; Xiaojia – 356666, 3439268) from satellite imagery and used these as the triggering locations for our simulations. We then prescribed three input volumes at each of these locations ($10^4 \ m^3, 10^5 \ m^3$ and $10^6 \ m^3$). The flow volumes simulate a range of observed post-2008 debris flows, representing low, high, and extreme debris flows documented in the Fan et al., (2019a) datasets. The volumes we selected reflects the range of similar hazard events in comparable geomorphological settings such as other parts of China and Italy (Wu et al., 2016; Bernard et al., 2019). For catchments with check dams, we added barriers at each check dam location by raising the cell count of the DEM by the height of the check dam obtained from field imagery.*

*The model was validated by comparing simulated runout extents with observed debris flows from post-2008 events. While a 30m resolution was the only available DEM for our study locations, we tested the sensitivity of DEM resolution on the extent of the final flow. A higher, 10m resolution DEM was available for the Cutou gully and we ran LAHARZ for that catchment. While the 10m DEM created a more effective flow path compared to the mapped data, the flow depositional area was similar in both the 10m and 30m scenario (RMSE 18m). Given the lack of a significant difference between the two DEM resolution we ran 30m scenarios across the three catchments. We note that there is not a strong understanding currently of what controls the maximum size of debris flows within Wenchuan catchments, hence we cannot attribute a particular probability to each scenario.*

2. **Could authors further explain how the 'levee effect' influence exposure of large-scale debris flow events?**

We have expanded the explanation of the 'levee effect' in the revised manuscript. This section now describes how levees can alter the natural flow patterns of debris, potentially redirecting flows or causing accumulations in areas that may not otherwise be exposed. We highlight Cutou specifically to demonstrate how the 'levee effect' influences debris flow risk in the context of our study area. This helps to better integrate the theory into our analysis of debris flow exposure and risks. Please see the additions below to be added to the revised manuscript:

*The levee effect can influence exposure to large-scale debris flow events by inadvertently increasing risk in areas protected by engineered mitigation structures, such as check dams. This occurs because the perceived safety provided by these structures can encourage development in vulnerable areas, which might otherwise remain uninhabited due to their high-risk nature. This phenomenon is best evidenced in our paper by the Cutou catchment, where the construction of check dams in 2013 coincided with widespread urban expansion, despite ongoing small-scale debris flow activity in the area. Subsequently, building exposure increased by 64% post-2008, underscoring the risk amplification associated with structural mitigation. This observation highlights the necessity of coupling structural interventions with strategies that address residual risks and foster community awareness of long-term hazard vulnerabilities. The 2019 debris flow event exemplified the risks associated with this effect, as the flow overtopped the check dams and used the stored material as a secondary fuel, significantly amplifying the impact. As a result, 40% of surveyed buildings were inundated, demonstrating how the levee effect can potentially escalate exposure to large-scale debris flow events.*

3. **Expand the analysis of the Xiaojia area to explore the specific reasons for its low exposure changes, such as natural terrain barriers, land-use planning, or building quality.**

We have included an expanded analysis of the Xiaojia area, focusing on the factors contributing to its relatively low exposure to debris flow events. Specifically, we noted factors of natural terrain barriers as well as land-use planning measures, including zoning and construction regulations that mitigate risk. Additionally, we have considered the quality of buildings in Xiaojia, which may influence the ability of structures to withstand debris flow events. These factors are now discussed in greater detail in the revised manuscript. Please see the additions below, in bold italics, to be added to the revised manuscript:

*Xiaojia was chosen as the comparative catchment due to the absence of engineered mitigation such as check dams. This analysis of Xiaojia therefore enables comparisons on the effectiveness and limitations of engineering approaches applied to Cutou and Chediguan. In Xiaojia, the lack of engineered dam structures, results in different erosion and deposition patterns compared to the other two catchments. Distinct patterns of upstream erosion and downstream deposition are observed, contrasting with the more controlled environments in the modified gullies, where deposition occurs on the northern channel flank and pronounced erosion on the southern flank. The data availability for building types, quality and spatial distribution was limited to remote sensing images and few literature sources, which restricts our ability to thoroughly assess how specific building characteristics, such as materials, influence the exposure of the built environment to debris flow hazard. This is particularly evident in Xiaojia, where more specific input data would be beneficial for understanding the role of urbanisation and construction practices on risk levels.*

*Our analysis of Xiaojia unveils no discernible relationship between building development and heightened exposure, particularly to residential and critical infrastructure. This lack of correlation is potentially linked to factors beyond simple urbanisation patterns, like construction quality, building regulations, presence of natural barriers, and effectiveness of mitigation measures. Natural terrain barriers observed in this gully including steep slopes and rocky outcrops, could limit the extent of debris flow impacts by reducing the mobility of debris and offering natural protection to certain areas. To fully understand this observation, further investigation into the above variables is warranted. The absence of significant urban expansion, particularly post-earthquake in Xiaojia may be a key factor in mitigating exposure. This area has experienced less intensive development compared to Cutou and Chediguan, where urban expansion following the implementation of check dams potentially increased exposure to debris flow hazards. Furthermore, the building quality in Xiaojia may play a significant role in*

*influencing its overall vulnerability. Without more detailed building-specific data, it is possible that buildings in Xiaojia may be of higher structural integrity or designed to withstand environmental stressors better than those in more developed catchments.*

*Additionally, detailed mapping of past debris flow events and their impacts on the built environment could provide insights into the specific mechanisms influencing vulnerability in Xiaojia. By conducting a more comprehensive analysis that considers these factors – especially in terms of land-use planning, construction standards and the role of natural terrain features at the local scale, we can gain a better understanding of the complex interactions between building development and exposure to natural hazards in Xiaojia. This, in turn, can inform more effective risk management and mitigation strategies tailored to the unique characteristics of the area. Development in Xiaojia primarily concentrates on the lower slopes (Fig 5(i) and (ii)) at the gully mouth, featuring the construction of major roads and highways (G213 and G2417), alongside the expansion of existing residential areas. Chediguan exhibits a less marked land cover transformation, owing to roads being directed through mountain tunnels. Notably, development in Xiaojia mainly surges post-earthquake up to 2010, with only minor construction activities documented thereafter (Fig 5(iii)).*

**4. The introduction and conclusion sections should better align with the research objectives.**

We have made edits to both the introduction and conclusion sections to better align with the research objectives. In the introduction, we now clearly outline the key research questions and objectives that guide the study. In the conclusion, we explicitly relate the findings back to the original research objectives, ensuring that the main contributions of the study are clearly communicated. This revision strengthens the coherence between the introduction, body, and conclusion of the manuscript, to ensure our study's main contributions are clearly communicated. Please see the changes to be added to the introduction of the revised manuscript in italics below to better align with research objectives:

*This study seeks to understand whether the addition of engineered mitigation measures, primarily check dams, have influenced the susceptibility of post-earthquake Wenchuan communities to large debris flows. We compare 3 catchments with similar topography and geology, but different levels of mitigation. We measure the building exposure in two neighbouring catchments with check dams (Cutou and Chediguan) and compare with a third, unmitigated gully (Xiaojia). We examine how infrastructure develops in the basins with time and as a function of check dam measures. By analysing infrastructure development in these catchments, particularly in Cutou and Chediguan in the years following mitigation – will seek to assess how check dam construction has impacted infrastructure growth and the potential exposure to debris flow events of different sizes. Additionally, our analysis will explore whether the presence of these structures has impacted risk perception and/or land-use decisions in 'at-risk' catchments.*

---

## Author Comment (AC2)

1. **Line 217 The facility value and related parameters selection were overlooked, and that increase the vagueness of application process in case the reader was interesting in similar application design.**

We have clarified and expanded our methodology section for assigning fragility values to buildings. To address the vagueness, we have explicitly stated that the fragility values were assigned based on a combination of literature sources and satellite images. Specifically, buildings that were inundated or damaged in previous events, or those located along the channel or gully mouth, were given a fragility value of 1, while all other buildings were assigned a value of 0. We validated these values using historical damage reports from the 2008 earthquake recovery period to ensure their applicability. Additionally, we emphasised this approach allows for replicable application designs in similar hazard-prone areas, thus addressing the concern regarding the transferability of the methodology to other contexts. We believe these revisions provide the necessary clarity for readers interested in applying this approach to similar designs. Please see additions for the revised manuscript, in italics, below:

*$E_b$ is the number of buildings damaged, and C is the fragility index of the elements at risk (Zou et al., 2019). Fragility values range from 0 to +1, with higher values indicating greater susceptibility to damage and/or failure. We assigned fragility values through using a mixture of literature and satellite images; buildings shown to be inundated or damaged in previous events or situated along the channel or gully mouth were given a value of 1, all other buildings were set a value of 0. These values were validated using historical damage reports, where available, from the 2008 earthquake recovery period to ensure applicability (Zeng et al., 2015; Wei et al., 2021; Petley et al., 2023). This approach allows for replicable application designs in similar hazard-prone areas.*

2. **The justification of using -1 to +2 as units of measure to be inserted and its quantification relationship to vulnerability value is missing.**

We have added a justification for using the -1 to +2 scale as units of measure in our analysis. The scale was selected based on its ability to represent a range of vulnerability values that are meaningful for our study area. We have also included a quantification relationship between this scale and vulnerability values, explaining how each unit on the scale corresponds to specific levels of vulnerability, both physical and economic. Please see the additions below, in bold italics, to be added to the revised manuscript:

*The key difference between our method and that of Zou et al (2019) is the incorporation a modification factor, M, to account for the effectiveness of engineered measures like check dams in mitigating building damage and subsequent exposure. The mitigation factor, M, quantifies the influence of engineered measures, in this study check dams, on the vulnerability and subsequent exposure of buildings to debris flow impacts. The addition of this factor brings an evaluative element to the exposure assessment, quantifying the influence of check dams and assigning values ranging from -1.0 to +2.0 to reflect a spectrum of mitigation outcomes:*

- *M = -1: Effective mitigation of debris flows, resulting in a significant reduction in hazard exposure, as evidenced by a decrease in the number of buildings damaged during historical events following construction.*
- *M = 0: No mitigation present; exposure levels are entirely dependent on natural site conditions.*
- *M = +1: Ineffective mitigation; there is no reduction in the number of buildings impacted in recorded debris flow events following dam construction.*

- *M = +2: Mitigation increases exposure. Recorded events of similar volume show an increase in the number of buildings impacted following dam construction.*

*The above -1 to +2 scale was selected to capture a nuanced relationship between mitigation effectiveness and vulnerability. A reduction in M (e.g., -1) lowers hazard exposure by reducing flow impacts at critical locations, thereby increasing $E_{df}$. Conversely, an increase in M (e.g., +2) elevates exposure, as development in hazard-prone areas amplifies the potential for damage. For example, a decrease in M by one unit (from 0 to -1) reflects an improvement in flow attenuation due to effective check dams, reducing overall exposure. Conversely, an increase in M by one unit (from 0 to +1) signifies a scenario where mitigation fails, e.g. the. 2019 debris flow event in Cutou, maintaining high exposure levels. At M = +2, exposure exceeds natural vulnerability due to increased hazard presence caused by intensified land use near mitigation structures.*

*This scale was developed through a combination of evaluating present hazard mitigation and analysing of historical data, particularly from the 2008 earthquake recovery. Moreover, this approach, based upon the methodology proposed by Zou et al. (2019), allows for an assessment of exposure by considering both the physical resistance of buildings and the efficacy of mitigation efforts.*

**3. LAHARZ simulation, data processing, assumption, and technical details were missing.**

We have now provided additional details regarding the LAHARZ simulation, including the specific data processing steps, key assumptions, and technical details. This includes a description of the model setup, the sources of input data, and the assumptions underlying the simulation parameters. We have also discussed the limitations of the model, and any uncertainties associated with the assumptions made. Please see additions for the revised manuscript, in italics, below:

*LAHARZ is a GIS toolkit for lahar hazard mapping and modelling, developed by the USGS to calculate the area of inundation and cross sections based on empirical scaling relationships between area and volume (Schilling., 2014; Iverson et al., 1998). These empirical relationships allow for the creation of realistic inundation areas without a priory knowledge of the rheological parameters. The model simulates a debris flow triggered at a source point located on a digital elevation model and with an initial source volume. The model calculates the flow path downslope of the triggering location then generates a cross-section at each point downslope that represents the depositional volume for that area (Iverson et al., 1998).*

*We implemented this model using the extension in ArcGIS (USGS., 2007). We used the 30m resolution ASTER DEM as an input, as it is the most reliable of the globally available DEMs. We identified the source areas of 2019 debris flows for Chediguan and Cutou and the 2011 for Xiaojia (Cutou – 351603, 3473449; Chediguan – 350846, 3453894; Xiaojia – 356666, 3439268) from satellite imagery and used these as the triggering locations for our simulations. We then prescribed three input volumes at each of these locations ($10^4\ m^3, 10^5\ m^3$ and $10^6\ m^3$). The flow volumes simulate a range of observed post-2008 debris flows, representing low, high, and extreme debris flows documented in the Fan et al., (2019a) datasets. The volumes we selected reflects the range of similar hazard events in comparable geomorphological settings such as other parts of China and Italy (Wu et al., 2016; Bernard et al., 2019). For catchments with check dams, we added barriers at each check dam location by raising the cell count of the DEM by the height of the check dam obtained from field imagery.*

*The model was validated by comparing simulated runout extents with observed debris flows from post-2008 events. While a 30m resolution was the only available DEM for our study locations, we tested the sensitivity of DEM resolution on the extent of the final flow. A higher, 10m resolution DEM was available*

*for the Cutou gully and we ran LAHARZ for that catchment. While the 10m DEM created a more effective flow path compared to the mapped data, the flow depositional area was similar in both the 10m and 30m scenario (RMSE 18m). Given the lack of a significant difference between the two DEM resolution we ran 30m scenarios across the three catchments. We note that there is not a strong understanding currently of what controls the maximum size of debris flows within Wenchuan catchments, hence we cannot attribute a particular probability to each scenario.*

**4. Fig. 7 and the amount of buildings, types, and degree of vulnerability in terms of economic or physical were missing.**

We have revised Figure 7 to include the total number of buildings, along with the degree of vulnerability. Additionally, the figure caption now provides this information for clarity. The lines below will be added to explain why we did not include economic data in the analysis:

*"However, due to the lack of available data on building materials in these three regions, we were unable to quantify their influence on structural vulnerability. As a result, exposure was determined to be the primary contributing factor to building damage."*

Revised Figure 7(i) with building numbers added and figure caption below (Figure 7(ii) will remain unchanged):

[Figure]

**Figure 7:** *Built environment impacts from three debris flow scenarios modelled using LAHARZ at Cutou, Chediguan and Xiaojia. (i). Percentage of buildings damaged as a proportion of total buildings (Cutou – 197, Chediguan – 69 and Xiaojia – 43) in each scenario. (ii) Total number of buildings damaged by each simulated debris flow.*

**5. Maps and figures are very simple, and the conclusion was almost predictable, as I am still looking for scientific arguments and proofs that may increase the credibility of research contribution.**

We appreciate the reviewer's feedback. In response, we have revised the introduction and research objectives to better highlight the key scientific arguments and methods used to substantiate the

credibility of our findings. These revisions aim to strengthen the scientific rigor and contribution of our research.

Regarding the maps and figures, we chose to keep them simple to ensure ease of interpretation and clear visualisation of exposure changes over time (Figures 5 and 6). We believe this approach enhances the accessibility of the findings without compromising the scientific integrity of the results.

---

## Author Response (AR1)

**Review 1:**

We would like to thank the reviewer for their insightful and constructive comments on our manuscript. The reviewer has highlighted several areas of the manuscript that required clarification, particularly regarding the explanation of the levee effect and the implementation of methodological parameters. In response to these suggestions, we have significantly expanded the methods section and added two new paragraphs to provide a clearer explanation of these analyses. Additionally, we have added a dedicated paragraph to explain the levee effect and adjusted the introduction and conclusion to ensure better alignment with our research objectives and improve overall clarity. Below, we provide specific responses to each of the reviewer's comments:

1. **Lines 143-226: The methodology section should include the rationale for the selection of model parameters and provide a sensitivity analysis to enhance the credibility of the model results.**
   We have revised the methodology section (lines 145-271) to include a detailed rationale for the selection of model parameters. This now explains the factors we considered when selecting each parameter including their relevance to the study area and impact on model accuracy. Furthermore, we have incorporated a sensitivity analysis of the DEM on the model's output. This addition enhances the robustness of the results and justify the model's applicability in the context of our research.

2. **Could authors further explain how the 'levee effect' influence exposure of large-scale debris flow events?**
   We have expanded the explanation of the 'levee effect' in the revised manuscript (lines 550-572). This section now describes how levees can alter the natural flow patterns of debris, potentially redirecting flows or causing accumulations in areas that may not otherwise be exposed. We highlight Cutou specifically to demonstrate how the 'levee effect' influences debris flow risk in the context of our study area. This helps to better integrate the theory into our analysis of debris flow exposure and risks.

3. **Expand the analysis of the Xiaojia area to explore the specific reasons for its low exposure changes, such as natural terrain barriers, land-use planning, or building quality.**
   We have included an expanded analysis of the Xiaojia area (lines 353 to 377), focusing on the factors contributing to its relatively low exposure to debris flow events. Specifically, we noted factors of natural terrain barriers as well as land-use planning measures, including zoning and construction regulations that mitigate risk. Additionally, we have considered the quality of buildings in Xiaojia, which may influence the ability of structures to withstand debris flow events. These factors are now discussed in greater detail in the revised manuscript.

4. **The introduction and conclusion sections should better align with the research objectives.**
   Response shown in lines 91 to 100 (introduction); Lines 612 to 648 (conclusion).

   We have made edits to both the introduction and conclusion sections to better align with the research objectives. In the introduction, we now clearly outline the key research questions and objectives that guide the study. In the conclusion, we explicitly relate the findings back to the original research objectives, ensuring that the main contributions of the study are clearly communicated. This revision strengthens the coherence between the introduction, body, and conclusion of the manuscript, to ensure our study's main contributions are clearly communicated.

**Review 2:**

We want to thank the Reviewer for their helpful comments. The reviewer has raised important points regarding the explanation of the LAHARZ model, the parameterisation of exposure analysis, and the

presentation of vulnerability data. In response, we have expanded several sections of the manuscript and edited Figure 7 to include additional details on building numbers and vulnerability levels. We have also worked to align our research objectives more clearly with our scientific arguments. Specific responses to each of the reviewer's comments are provided below:

1. **Line 217 The facility value and related parameters selection were overlooked, and that increase the vagueness of application process in case the reader was interesting in similar application design.**
   We have clarified and expanded our methodology section for assigning fragility values to buildings (lines 233 to 239) . To address the vagueness, we have explicitly stated that the fragility values were assigned based on a combination of literature sources and satellite images. Specifically, buildings that were inundated or damaged in previous events, or those located along the channel or gully mouth, were given a fragility value of 1, while all other buildings were assigned a value of 0. We validated these values using historical damage reports from the 2008 earthquake recovery period to ensure their applicability. Additionally, we emphasised this approach allows for replicable application designs in similar hazard-prone areas, thus addressing the concern regarding the transferability of the methodology to other contexts. We believe these revisions provide the necessary clarity for readers interested in applying this approach to similar designs.

2. **The justification of using -1 to +2 as units of measure to be inserted and its quantification relationship to vulnerability value is missing.**
   We have added a justification for using the -1 to +2 scale as units of measure in our analysis (lines 243 to 264). The scale was selected based on its ability to represent a range of vulnerability values that are meaningful for our study area. We have also included a quantification relationship between this scale and vulnerability values, explaining how each unit on the scale corresponds to specific levels of vulnerability, both physical and economic.

3. **LAHARZ simulation, data processing, assumption, and technical details were missing.**
   We have now provided additional details regarding the LAHARZ simulation (lines 200 to 218), including the specific data processing steps, key assumptions, and technical details. This includes a description of the model setup, the sources of input data, and the assumptions underlying the simulation parameters. We have also discussed the limitations of the model, and any uncertainties associated with the assumptions made.

4. **Fig. 7 and the amount of buildings, types, and degree of vulnerability in terms of economic or physical were missing.**
   We have revised Figure 7 to include the total number of buildings, along with the total number of catchment buildings added to the figure caption. Additionally, the figure caption now provides this information for clarity. In lines 566 to 570, we explain why we did not include economic data in the analysis.

5. **Maps and figures are very simple, and the conclusion was almost predictable, as I am still looking for scientific arguments and proofs that may increase the credibility of research contribution.**
   We appreciate the reviewer's feedback. In response, we have revised the introduction and research objectives to better highlight the key scientific arguments and methods used to substantiate the credibility of our findings (lines 94 to 101). These revisions aim to strengthen the scientific rigor and contribution of our research. Regarding the maps and figures, we chose to keep them simple to ensure ease of interpretation and clear visualisation of exposure changes over time (Figures 5 and 6). We believe this approach enhances the accessibility of the findings without compromising the scientific integrity of the results.

---

## Referee Report (RR1)

Since I'm a late addition to the review here, in fairness to the authors I've tried to primarily focus on how the authors addressed concerns from the previous stage. I'm currently stuck in a cycle of receiving a new set of concerns from a new set of reviewers repeatedly, and I don't want to do the same thing to someone else! In general, I think the authors have done a pretty good job of addressing prior reviewers' concerns in the previous round of revision. I'll raise some issues in the line-by-line comments that are mostly related to clarity and presentation and could be addressed without any new analysis. The biggest issue I see scientifically is the centrality of the argument about the levee effect being driven by check dam construction – this is an important thing to get right for work like this that could potentially influence decision making and I don't think that a dataset of development 3 catchments, two with check dams and one without is compelling support for the levee effect here. However, there's still some interesting stuff here that should be more or less ready to publish with some reframing and clarification.

28-30 – incomplete sentence – needs copy edit

107-108 – Please clarify this sentence – it sounds like landslides are being included as a subcategory of debris flows as written. Also – why would debris flows be constrained to the south? I can find lots of documentation about debris flows in northern provinces in China, maybe eliminate.

179 – Clarify "properties"

190 – commas make this sentence awkward

231 – Is this assessed cell by cell?

236 – Why is fragility just assessed as a binary? Would be nice to have a little clarification there.

Figure 3 – What do the numbered tributaries correspond to?

362-363 – This is a little confusing per earlier – isn't exposure just being assessed by whether a structure is in the path of a debris flow or not?

Figure 5 – In Cutou Gully, the shading of the tributary channel with the pale blue color only implies that it wasn't there 2005-2010? Also, some of the roads (dashed purple, solid red) lack symbols in the legend. Not a huge deal though. Also, were any of the areas shaded as built environment in 2005-2010 subject to debris flows? It's a little hard to parse whether the

changes in built environment from 2011-2019 represent just additions to pre-existing infrastructure or replacement to areas that were damaged. More importantly, the spatial scale is not correct. The small scale bar seems to represent 5 km (it's pretty small, I may be misreading) but checking out the field sites on Google Earth they're an order of magnitude smaller than that.

383-390 – Given the questions I had about figure 5, maybe it would be helpful to include some example satellite images showing damage in these events and your methodology of assessment. This would be fine just as a supplement too.

397-398 – Is there an available volume estimate for any of the debris flows in these catchments? Forgive me if you already listed it somewhere and I just couldn't find it.

Figure 6 – By "low and high runouts" do you mean the 10^4 and 10^5 m^3 debris flow simulations in LAHARZ? Please clarify. Maybe it would be helpful to show the result of some of the smaller simulations at a smaller scale, though, to visualize the effect of the check dams.

412-415 – Please clarify what we're looking at in 6iii that shows the effectiveness of the check dams – I'm struggling to see what's illustrative in the figure. Also, the confusion I had a little earlier with visualizing expansion in the built environment in Figure 5 is compounded here – the text states that expansion was restrained in Xiaojia Gully but in Figure 6iii it appears that all the development was post-2005 at the very least.

415-417 – I agree that the levee effect might be an important consideration here and the check dams might have played a role in encouraging development, but three catchments is an extremely small sample size to rely on. I hesitate to ask the authors to expand on their analysis at this point, but it would substantially strengthen the paper to look at a few other catchments with and without check dams in place and see whether this trend holds up to a larger sample size.

420-423 – I don't quite understand what this means. The landscape seems like a really important factor in all these simulations, but I wouldn't invoke it if you're not directly analyzing it here – it just gets the reader thinking that it might be more important than the check dams.

Figure 7 – What drives the extreme jump up in destruction in Cutou under the largest debris flow volume simulation? Also, I think it would be more effective to have at least a little analysis of debris flow inundation decoupled from building damage since that's dependent on a lot of

factors not controlled for in your analysis. How do the check dams control the distribution of erosion and deposition?

469 – I don't think this even needs to be stated

481-482 – How did the dams fail? Collapse? Overtopping?

519-520 – Declined from what? Earlier industrial presence? Or is it just less industrialized than the other catchments?

525-526 – Doesn't it go without saying that a larger debris flow is going to subject a larger area to potential damage?

529-530 – is it that check dams are not effective against the largest flows or that the check dams just aren't big enough to be effective?

531 – What results if you simulate large debris flows without the check dams present?

532 – "raised development level" to me implies an increase in elevation – maybe reword.

551 – Needs more than 3 data points for substantive support of this claim, I'd say. A lot of other factors could be encouraging or discouraging development.

566 – Could it also be that construction of check dams has been prioritized in areas with a lot of development?